# Canonical Hedgehog Pathway and Noncanonical GLI Transcription Factor Activation in Cancer

**DOI:** 10.3390/cells11162523

**Published:** 2022-08-14

**Authors:** Chamey Suchors, James Kim

**Affiliations:** 1Nancy B. and Jake L. Hamon Center for Therapeutic Oncology Research, University of Texas Southwestern, Dallas, TX 75208, USA; 2Department of Internal Medicine, Division of Hematology-Oncology, University of Texas Southwestern, Dallas, TX 75208, USA; 3Harold C. Simmons Comprehensive Cancer Center, University of Texas Southwestern, Dallas, TX 75208, USA

**Keywords:** Hedgehog signaling pathway, cancer, noncanonical GLI activation

## Abstract

The Hedgehog signaling pathway is one of the fundamental pathways required for development and regulation of postnatal regeneration in a variety of tissues. The pathway has also been associated with cancers since the identification of a mutation in one of its components, *PTCH*, as the cause of Basal Cell Nevus Syndrome, which is associated with several cancers. Our understanding of the pathway in tumorigenesis has expanded greatly since that initial discovery over two decades ago. The pathway has tumor-suppressive and oncogenic functions depending on the context of the cancer. Furthermore, noncanonical activation of GLI transcription factors has been reported in a number of tumor types. Here, we review the roles of canonical Hedgehog signaling pathway and noncanonical GLI activation in cancers, particularly epithelial cancers, and discuss an emerging concept of the distinct outcomes that these modes have on cancer initiation and progression.

## 1. Introduction

Originally discovered as a singular secreted protein during *Drosophila* development, the mammalian Hedgehog (Hh) homolog consists of three ligands, Sonic Hedgehog (SHH), Indian Hedgehog (IHH), and Desert Hedgehog (DHH) [1,2,3,4]. Each of these ligands plays a crucial role in fundamental developmental processes, including cell proliferation, differentiation, and survival [5]. Mutations that induce dysregulated Hh signaling can produce a variety of congenital malformations including holoprosencephaly (HPE), Greig cephalopolysyndactyly syndrome (GCPS), and Pallister–Hall Syndrome (PHS) [6,7,8,9,10]. Postnatally, Hh pathway activity continues to regulate stem cell maintenance and tissue homeostasis [11]. However, aberrant activation of the pathway has been linked to numerous cancers, as was first noted in patients with Basal Cell Nevus Syndrome (or Gorlin Syndrome, BCNS)—a hereditary condition associated with predisposition for basal cell carcinoma, medulloblastoma, and rhabdomyoma [12,13,14]. Subsequent reports have implicated Hh pathway activity in a large variety of other cancers, including those of the brain, liver, lung, pancreas, stomach, breast, colon, gallbladder, prostate, and hematological malignancies [15,16].

The abnormal activity of the Hh signaling pathway in mammalian cells is propagated via two principal modes—canonical and noncanonical activation. The canonical Hh pathway is that which proceeds via the developmental signaling cascade, i.e., from a Hh ligand to Patched (PTCH) to Smoothened (SMO) to GLI transcription factors. Mutations in pathway components downstream of Hh ligands can lead to ligand-independent canonical activation of the pathway. In contrast, noncanonical activation of the Hh signaling pathway refers to modes in which only some of the pathway components are utilized, including PTCH-dependent, SMO-independent signaling, SMO-dependent, GLI-independent signaling and SMO-independent activation of GLI transcription factors. The noncanonical activation of GLI transcription factors constitute the majority of the reports of noncanonical pathway activation in cancers. Reports of the other modes of noncanonical pathway activity have been described primarily in benign and developmental contexts with little in cancers. Therefore, we will focus on Hh ligand-independent, SMO-independent activation of GLI transcription factors in our discussion of noncanonical activity of the pathway in cancers. Noncanonical activation of GLI1 and GLI2 often occurs through crosstalk with other pathways driven by oncogenic drivers or loss of tumor suppressors [5].

Here, we will review the literature for canonical Hh pathway and noncanonical GLI activation in tumor development and growth and present an emerging model of the roles of the pathway in tumor epithelia and tumor stroma.

## 2. Hedgehog Signaling Pathway

A simplified schematic of the Hh signaling pathway is presented in Figure 1. In the absence of a Hh ligand, Patched (PTCH), a 12-pass transmembrane protein, inhibits Smoothened (SMO), a 7-pass transmembrane protein, in a non-stoichiometric manner [17] via the export of endogenous sterols from the inner leaflet of the plasma membrane to deprive SMO of the sterols necessary for its activation [18,19,20,21,22,23]. Upon binding of a Hh ligand to 2 PTCH molecules, the N-terminal palmitate of the Hh ligand is inserted into the sterol tunnel of 1 PTCH molecule to block the export of sterols. The binding of the Hh ligand to a second PTCH molecule induces endocytosis of the complex away from the primary cilia [21,22,24,25]. Phosphorylation of SMO cytoplasmic tail by casein kinase 1α (CK1α) and GPCR kinase 2 (GRK2) induces an active conformation of SMO in mammalian cells [26] although SMO with mutations in the phosphorylation sites still retained activity in zebrafish embryos [27]. Ultimately, SMO is localized to the primary cilium via interactions with β-arrestin [28,29], where SMO is able to interact with accessible cholesterol and trigger a signal cascade that results in the transcription of Hh target genes by GLI transcription factors.

When SMO is inactive, Suppressor of Fused (SUFU) binds to the GLI transcription factors and prevents their translocation to the nucleus [30]. Protein kinase A (PKA), casein kinase 1 (CK1), and glycogen synthase kinase 3β (GSK3β) subsequently phosphorylate the GLI transcription factors to mark them for proteolysis [31,32,33]. The transcription factors, GLI2 and GLI3, can be proteolytically truncated into a repressive form. GLI3 is most commonly in its repressor form (GLI3R) as it is efficiently processed and translocated to the nucleus to inhibit transcription of Hh target genes. GLI2 repressor (GLI2R) is significantly less stable than GLI3R and is rapidly degraded [34]. Thus, GLI3 resides primarily as a transcriptional repressor whereas GLI2 is primarily a transcriptional activator.

With the binding of a Hh ligand to PTCH and the activation of SMO, SUFU bound to GLI traffics to the primary cilium [35,36,37] and GLI1/2 dissociates from SUFU with the aid of Ellis van Creveld syndrome proteins, EVC and EVC2 [38,39,40]. Sequestration of PKA by SMO at the membrane prevents PKA-mediated phosphorylation of GLI proteins and subsequent degradation [41]. Full length GLI2, the primary activating GLI transcription factor, then translocates to the nucleus to begin transcription of Hh pathway target genes including *GLI1*, *PTCH1*, and *HHIP*. Transcription of these target genes are commonly utilized as reporters of pathway activity.

In mammals, there are three GLI transcription factors, GLI1, GLI2, and GLI3, in contrast to the homologous singular Cubitus interruptus (Ci) in *Drosophila* [42,43]. All three GLI factors have a highly conserved zinc-finger motif in their DNA-binding domains which target the 9-mer sequence GACCACCCA [44]. However, unique N-terminal and C-terminal domains of the three GLI transcription factors determine their individual function. All three GLI factors possess an activator domain in their C-termini; however, the N-termini of GLI2 and GLI3 harbor a repressor domain which is not present in GLI1. Therefore, GLI2 and GLI3 are able to be transformed into repressor forms (GLIR) by proteolytic deletion of their C-termini, although GLI3R acts primarily as a transcriptional repressor, and GLI2 primarily acts as a transcriptional activator, as noted previously [43].

## 3. Modes of Canonical Hedgehog Pathway Activation in Cancers

Four modes of canonical pathway activation in cancers have been proposed: (1) cell autonomous Hh ligand-independent activation due to mutations in pathway components, (2) cell autonomous Hh ligand-dependent autocrine activation, (3) non-cell autonomous paracrine activation, and (4) non-cell autonomous reverse paracrine activation [15]. Mutations in components of the canonical Hh signaling pathway, such as *PTCH* or *SMO*, can lead to aberrant loss or activation of pathway activity. Such mutations during development can lead to embryonic lethality or developmental disorders. Postnatally, mutations of Hh pathway components that induce pathway activation can engender cancers and will be discussed later. Ligand-dependent autocrine Hh signaling occurs when tumors cells secrete Hh ligands that bind to PTCH on the same cell and activates the pathway via activation of SMO. Early studies of the pathway in sporadic epithelial cancers utilizing cancer cell lines in cell culture conditions and mouse xenografts suggested that an autocrine mechanism was dominant. However, subsequent studies suggested that a paracrine mechanism may be dominant which will be discussed later in this review. In ligand-dependent paracrine Hh pathway signaling, tumor cells recapitulate development by expressing a Hh ligand that acts on neighboring stromal cells. In response, stromal cells secrete a variety of factors to affect the proliferation, differentiation, and survival of tumor cells. In the ligand-dependent reverse paracrine Hh signaling model, Hh ligands are secreted from the surrounding stromal cells and activate the pathway in tumor cells. Such a mechanism has been reported primarily in lymphomas and multiple myeloma [45]. A summary of cancers with canonical activation of the Hh signaling pathway is listed in Table 1.

## 4. Cancers with Ligand-Independent Canonical Pathway Activation

### 4.1. Basal Cell Carcinoma

Basal cell carcinoma (BCC) is the most common cancer in the U.S. and originates from the basal layer of the epidermis [81]. Most BCC arise from mutations in components of the Hh pathway, primarily *PTCH1* with ~10% in *SMO* [46,47]. The causative genetic mutation for BCC in a Basal Cell Nevus Syndrome (BCNS) patient was first mapped to chromosome 9q22 [12] and subsequently identified to *PTCH1* [14,48]. The mutations inactivate PTCH1 and, thus, allow SMO to be activated to initiate the signaling cascade of the Hh pathway. The discovery of this clear connection between biallelic loss of function *PTCH1* mutations and BCC development was the first of many links to be made between aberrant Hh pathway activity and cancers [15].

Genetic aberrations of pathway components have also been demonstrated both in mouse models and sporadic human BCCs that develop independently of an underlying genetic disorder. Mice overexpressing GLI2 display developmental defects in multiple organ systems and also generate tumors with a strikingly similar histology to human BCC [82]. Mice overexpressing GLI1 have also been shown to spontaneously generate BCC tumors by 10 weeks of age, even with intact p53 expression [83]. A variety of spontaneous, conventional, and conditional *Ptch* knockout mice that develop BCCs have been described [84]. Of note, the mutant *Ptch* mouse models require ultraviolet or ionizing radiation to induce BCC formation [84]. In addition to BCNS-connected basal cell carcinomas, sporadic BCCs have also been linked to ligand-independent Hh pathway activity. Both activating *SMO* mutations [49] and inactivating *PTCH* mutations [13,50] have been found in patients with sporadic BCC.

As *PTCH1* mutations are dominant in BCC, the activity of several SMO antagonists has been investigated in clinical trials [85]. In several phase II trials of locally advanced (laBCC) and metastatic BCC (mBCC), vismodegib (GDC-0449, [86]) had overall response rates of 60–69% and 37–49% for sporadic laBCC and mBCC, respectively [87,88,89]. Vismodegib was effective in reducing BCC tumor burden and inhibiting new BCC tumor formation in BCNS patients [90]. Sonidegib (LDE225, [91]), another small molecule SMO antagonist, had overall response rates of 56% and 8% for laBCC and mBCC, respectively, at 200 mg daily dosing and 46% and 17% for laBCC and mBCC, respectively, at 800 mg daily dosing. However, the 800 mg dosing schedule was associated with increased rates of adverse events leading to dose interruptions, reductions, and discontinuations compared to the 200 mg dosing schedule [92]. Vismodegib has been granted FDA approval for use in laBCC treatment and mBCC [93], while sonidegib has been granted FDA approval for laBCCs that are not amenable to surgery or radiotherapy [94].

Treatment resistance to vismodegib has arisen in BCC patients. Resistance mechanisms identified thus far in BCC include genetic alterations in Hh pathway components and bypass activation of GLI transcription factors. Bypass activation of GLI will be discussed later in this review, and we will focus here on genetic alterations of pathway components, in particular, drug-resistant SMO mutations. Two studies performed whole exome sequencing of vismodegib-resistant human BCC [95,96]. They identified genetic alterations in Hh pathway components downstream of PTCH in the vast majority of vismodegib-resistant samples including *SMO*, *SUFU*, *GLI2,* and *GLI3* mutations, and *GLI2* amplifications among others. Among these alterations, *SMO* mutations occurred most frequently and was found in 69–77% of the samples. Two types of *SMO* mutations were identified: mutations within or adjacent to the binding pocket that interfered with drug binding (including D473 mutant identified in a drug-resistant medulloblastoma patient [97]) and mutations outside the binding pocket, many of which confer constitutive activity to SMO (including the oncogenic W535L mutation identified in BCC [49,98,99]). Analogous to resistance mechanisms to small-targeted therapies against *EGFR* mutant lung cancer and *BCR-ABL* chronic myelogenous leukemia, the most common mechanism of vismodegib resistance occurs in its therapeutic target, SMO.

### 4.2. Medulloblastoma

Medulloblastoma is a cancer of cerebellar progenitor cells that is the most common brain malignancy in children but can also occur in adults [100]. Medulloblastoma is categorized into four distinct molecular subgroups based on transcriptomic data [101,102,103,104]. Hh pathway activity drives one of these groups, denoted as the “SHH group”, and correlates with desmoplastic histology. The SHH group of medulloblastoma accounts for ~30% of all medulloblastoma diagnoses, has an intermediate prognosis [105], and follows a bimodal distribution of incidence—a first peak in infants and young children less than 5 years old and a second peak in older adolescents and adults [106]. During cerebellar development, SHH ligand is secreted from Purkinje neurons and activates the pathway in lineage-restricted cerebellar granule neuron precursor cells to induce proliferation of the cerebellar precursor cells [107,108,109]. Aberrant activation of the Hh signaling pathway can lead to development of medulloblastoma. Mice with loss of function mutations in PTCH with or without loss of another tumor suppressor [51,52,53], SUFU with p53 loss [54], and activating mutations in SMO [55] develop medulloblastoma. In human medulloblastoma, mutations in *PTCH1*, *SMO*, and *SUFU*, and amplifications of *SHH*, *GLI2*, and *MYCN*, a pathway target gene, have been identified [56,57,58].

Both sonidegib and vismodegib have been tested in phase I and II clinical trials for their efficacy and toxicity against both adult and pediatric recurrent or treatment refractory medulloblastoma [110,111,112,113,114,115]. A meta-analysis of 5 early stage clinical trials demonstrated that neither drug was effective against non-SHH type medulloblastomas [110]. Vismodegib lengthened progression-free survival in recurrent SHH-subgroup medulloblastoma, while also maintaining a low toxicity profile [114]. Sonidegib showed similar antitumor effects in recurrent SHH-subgroup medulloblastoma [115], although sonidegib may have been more effective than vismodegib in pediatric medulloblastoma [110].

Relevant to use of SMO antagonists in children, mice treated transiently with HhAntag (or Hh-Antag691), a SMO antagonist [116], developed on target toxicity of permanent growth plate fusion of bones resulting in bone shortening and abnormal joints [117]. Two of the early clinical studies [112,114] did not find any dental or skeletal toxicities in skeletally premature children treated with vismodegib. However, in a phase I/II study of sonidegib in a pediatric population [115], three children were found to have focal growth plate closures. Furthermore, two children enrolled in the ongoing first line SJMB12 trial (NCT01878617) and one child off study with SHH group medulloblastoma were reported to develop widespread growth plate fusions, significant short stature and precocious puberty [118]. All of these children were skeletally premature (females <15 years old, males <17 years old) and had prolonged exposure to the SMO antagonists (>140 days), except for one child with 57 days of sonidegib exposure. Based on these findings, the protocol for SJMB12 trial has been modified such that only skeletally mature patients may receive vismodegib on study [115].

### 4.3. Rhabdomyosarcoma/Rhabdomyoma

Aberrations in muscle progenitor cell growth and differentiation can lead to benign rhabdomyoma and malignant rhabdomyosarcoma. Rhabdomyosarcoma is the most common soft tissue sarcoma in pediatric patients [119]. The connection between the Hh signaling pathway and rhabdomyosarcoma and rhabdomyoma was first identified in BCNS patients [120]. Retrospective and human cell line studies of rhabdomyomas and rhabdomyosarcomas noted increased expression of PTCH1 and GLI1, suggesting activation of the Hh pathway [59,121,122,123,124]. The presence of genetic aberrations in Hh pathway components in rhabdomyosarcoma is controversial, with studies suggesting the presence of such mutations [59,60] and studies that did not find any mutations in pathway components [122,125,126].

Rhabdomyosarcomas occurred in all *CAGGS-CreER*; *R26-SmoM2* mice after postnatal administration of tamoxifen [55]. The mouse model expresses the constitutively active mutant SMO-M2 ubiquitously after tamoxifen administration. A closer examination of the murine rhabdomyosarcomas from the *CAGGS-CreER*; *R26-SmoM2* mice suggested that they more closely resembled fetal rhabdomyoma histologically than aggressive rhabdomyosarcomas with upregulation of *Gli1* and *Ptch1* mRNA transcription, indicative of Hh pathway activation [61]. Mutations in *PTCH1* and *SMO* as well as homozygous deletions in *PTCH1* were identified in three out of five human fetal rhabdomyoma samples [61].

### 4.4. Ameloblastoma

Ameloblastoma is a rare, locally aggressive jaw tumor that rarely metastasizes but has a high risk of recurrence [127]. SHH is secreted from the tooth epithelium to activate the pathway in mesenchyme and pathway activation is critical for tooth development [128,129]. *PTCH1* mutations have been identified in odontogenic keratocysts of BCNS patients and sporadic cases [130]. A number of studies have identified protein or mRNA expression of Hh pathway components in ameloblastoma [131,132,133,134]. More recently, *SMO* mutations were identified in 39% of 28 ameloblastoma samples, primarily from maxillary ameloblastoma that tended to recur early [62]. All but one of the mutations generated SMO L412F mutations, with W535L mutation being the other mutation. Both mutations endow constitutive activity to SMO and are resistant to vismodegib [62].

### 4.5. Meningioma

Meningiomas are the most common intracranial tumor [135]. Up to 80% of the tumors are benign (WHO Grade I), whereas the other 20% tend to be more aggressive and recur after standard of care resection [135]. From a molecular perspective, meningiomas are divided between NF2 mutant and non-mutant meningioma. NF2 mutant meningiomas account for approximately 60% of all meningiomas [63]. Among the non-NF2 mutant meningioma subgroup, mutations in *SMO* and *SUFU* were identified [63,64,65]. SMO L412F and W535L mutations occur in approximately 5% meningiomas [64,65]. SUFU R123C mutant was identified in <1% of meningioma [66].

## 5. Cancers with Ligand-Dependent Canonical Hedgehog Pathway Activation in Cancer

### 5.1. Tumor-Suppressive Action of Ligand-Dependent Paracrine Canonical Hh Signaling Pathway

A growing body of literature over the past decade in preclinical models of endoderm-derived epithelial cancers have suggested that canonical paracrine activation of the pathway in stromal cells suppress tumor growth and progression. Genetic loss or pharmacologic inhibition of the pathway in stroma led to accelerated tumor growth and metastases [136]. The lack of positive outcomes from a number of clinical trials with SMO antagonists further support these preclinical results.

#### 5.1.1. Bladder Cancer

Urothelial cancer accounts for >75% of bladder cancers with tumor grading based on the level of infiltration into the bladder wall muscle, with non-invasive urothelial carcinomas being assigned a lower grade [137]. Interest in the relationship between Hh signaling pathway and bladder cancer arose in the late 1990s when urothelial carcinomas were determined to have loss of heterozygosity deletions (LOH) on chromosome 9q, which includes the region encoding for *PTCH1*, hinting that the Hh pathway may support urothelial carcinoma proliferation [138,139,140,141]. Genomic sequencing of 408 muscle invasive bladder cancer samples as part of The Cancer Genome Atlas project [142] found genetic alterations in *PTCH1*, *SMO*, and *SUFU* in 5%, 3%, and 2.7% of samples, respectively, suggesting that ligand-independent canonical pathway activity may be important in ~10% of tumor invasive bladder cancers.

Bladder cancer was the first cancer to report that canonical paracrine Hh pathway in stroma suppressed tumor growth. Injured bladder urothelial cells induce secretion of SHH from the basal epithelia to activate the pathway in stroma that, in turn, secretes WNT2, WNT4, and FGF16 to induce urothelial proliferation and repair [143]. Furthermore, SHH-expressing urothelial basal cells were identified as cells of origin for urothelial carcinoma, but SHH expression was ultimately lost in urothelial carcinomas [73]. Corroborating these findings, *SHH* mRNA was highly expressed in 96% of non-muscle invasive bladder cancers but decreased to 51% of the more aggressive muscle invasive bladder cancers suggesting the loss of Hh pathway activity with increasing tumor aggressiveness [144]. In an autochthonous murine model of N-butyl-N-4-hydroxybutyl nitrosamine (BBN) carcinogen-induced urothelial cancer, genetic loss of *Smo* in GLI1-expressing stromal cells led to loss of pathway activation in stromal cells, accelerated tumor growth, decreased survival, and loss of the differentiation factors, BMP4 and BMP5, in stromal cells [74]. Treatment of BBBN-treated mice with low dose FK506 to stimulate BMP pathway activity led to inhibition of tumor formation suggesting that loss of stromal Hh pathway activity early in the tumorigenic process leads to loss of BMP differentiation factors that restrain aggressive tumor formation [74].

#### 5.1.2. Pancreas Cancer

Despite having a relatively low incidence rate, pancreatic ductal adenocarcinoma (PDAC) has one of the highest mortality rates amongst all cancers, largely due to a dearth of effective treatments for advanced disease [145].

The precise relationship between the Hh signaling pathway and PDAC has been controversial. Mutations in Hh pathway components are rare [146,147] suggesting that intact canonical ligand-dependent pathway was active in pancreas cancers. Early studies reported the oncogenic roles of the Hh pathway in pancreas cancer cell lines in cell culture and in vivo [148,149,150,151] studies via paracrine interactions with stroma [150,151] and in an autochthonous mouse model of PDAC [152]. Based on these reports, several therapeutic clinical trials with SMO antagonists were conducted. However, the results of these trials were either equivocal or worse when compared to cytotoxic chemotherapies and, thus, raised questions regarding the oncogenic role of the pathway in PDAC [153,154,155].

Two preclinical studies re-examined the role of paracrine Hh pathway activity in stroma utilizing autochthonous mouse models of PDAC [75,76]. Loss of SHH expression in endogenous PDAC cells of *Pdx1-Cre*; *Kras^LSL-G12D/+^*; *p53^fl/fl^*; *Rosa26^LSL-YFP/+^*; *Shh^fl/fl^* mice led to worse survival, accelerated tumor growth, a more undifferentiated tumor histology with elevated expression of epithelial to mesenchymal transition (EMT) markers (ZEB1, SLUG), and it increased metastasis compared to the corresponding *Shh^WT^* mice [75]. Pharmacologic inhibition of the Hh signaling pathway with the SMO antagonist, IPI-926 [156], in *Pdx1-Cre; Kras^LSL-G12D/+^*; *p53^fl/fl^* mice also worsened survival. SHH activated the pathway in pancreatic fibroblasts and loss of SHH resulted in a decreased number of tumor fibroblasts. Similar results were reported in other autochthonous murine models of PDAC with SHH loss, *Ptf1a-Cre; Kras^G12D^*; *Shh^fl/fl^* and *Ptf1a-Cre*; *Kras^G12D^*; *p53^fl/fl^*; *Shh^fl/fl^* [76]. *Kras^LSL-G12D/+^*; *Ink4a/Arf^−/−^*; *Pdx1-cre*; *Gli1^nLacZ/+^* mice treated with cerulein, a cholecystokinin analogue that induces pancreatitis and accelerates preneoplastic pancreatic intraductal neoplasia (PanIN), followed by SMO antagonism with vismodegib, which accelerated PanIN formation and decreased fibroblasts and desmoplastic stroma. Conversely, treatment of the same mouse model with the SMO agonist, SAG21k [157], diminished PanIN formation and increased the number of fibroblasts and desmoplastic stroma [76]. Corroborating the tumor-suppressive function of fibroblasts in pancreatic cancer, loss of pancreatic fibroblasts in *Pft1a^cre/+^*; *Kras^LSL-G12D/+^*; *Tgfbr2^fl/fl^*; *aSMA-tk* mice that develop PDAC while losing myofibroblasts when treated with ganciclovir showed accelerated tumor growth, more undifferentiated tumor histology, and worse survival [158]. Taken together, paracrine activation of fibroblast Hh pathway by SHH acts as a tumor suppressor to inhibit pancreatic tumor growth.

#### 5.1.3. Colon Cancer

The importance of paracrine Hedgehog signaling in intestinal development has long been established with IHH, in particular [159]; however, the role of the pathway in colorectal cancers is debated. An early study detected mRNA transcripts of *SHH*, *IHH* in all 11 human colorectal cell lines but detected G*LI1* mRNA transcripts in only a third of the cell lines and none with *PTCH1* mRNA, raising some doubt as to whether the Hh signaling pathway is involved in colorectal cancer [149]. A subsequent study reported the stromal activation of the pathway by Hh ligands secreted from human colorectal cell lines [150]. Treatment with the SMO antagonist, HhAntag, inhibited the growth of flank xenografts in vivo [150]. A phase II clinical trial tested chemotherapy and bevacizumab (an anti-VEGA antibody) with or without vismodegib in metastatic colorectal cancer patients; the trial found no response or survival benefit with the addition of vismodegib nor did expression of *SMO*, *GLI1*, or *PTCH1* mRNA in tumor tissue predict a response [160].

In a carcinogen/chemical colitis murine model of colon cancer utilizing azoxymethane (AOM) and dextran sodium sulphate (DSS), Hh pathway activity was downregulated in the tumors and surrounding stroma [77]. Deletion of *Ihh*, in intestinal epithelial followed by AOM/DSS, led to loss of intestinal stromal pathway activity and increased tumor burden. Conversely, increase in stromal pathway activity with partial loss of *Ptch1* in stroma led to a decrease in stromal BMP inhibitors and decreased tumor burden [77]. Inhibition of stromal Hh pathway activity with genetic loss of *Smo* or treatment with vismodegib in the AOM/DSS model increased tumor burden. Furthermore, loss of stromal pathway by genetic *Smo* deletion or treatment with XL-139 (a SMO antagonist) worsened DSS-induced inflammation and colitis [78]. Conversely, activation of the pathway in stroma by partial *Ptch1* loss or treatment with the SMO agonist SAG21k [157] ameliorated the DSS-induced colitis with increased IL-10 expression and increased CD4+ Foxp3+ regulatory T cells [78].

#### 5.1.4. Prostate Cancer

Hedgehog signaling plays a critical role in the development and maintenance of the prostate. Pathway activity is not necessary for induction of the prostate but critical for its growth, proliferation of ductal tip epithelia, and suppression of ductal tip number [161]. In the adult prostate, loss of Hh pathway activity in prostate stroma increased epithelial tubule branching via indirect upregulation of HGF that was mediated by the microRNAs, miR-26a and miR-26b [162]. Moreover, loss of the androgen receptor in Hh-responsive GLI1-expressing stromal cells during embryogenesis disrupted prostate development, whereas loss of AR in stromal cells during prepubescence significantly inhibited prostate growth and regeneration [163].

Early studies of Hh pathway activity utilizing prostate cancer cell lines in cell culture and in vivo xenograft experiments suggested that the pathway had a tumor supportive role [164,165,166]. However, in the *CAGGSCreER; R26-SmoM2* mouse model with ubiquitous expression of SMO-M2 including the prostate epithelium, no cancerous morphology was detected in the prostate, even after 12 months [55], suggesting that constitutive Hh pathway activity was insufficient for prostate tumorigenesis. In a conditional prostate cancer mouse model in which MYC is expressed under the probasin promoter (PB-MYC), the tumors were highly representative of human prostate cancers, expressed high levels of IHH, and smooth muscle cells were depleted [79]. The study utilized a *Gli1^CreER/+^*; *R26^LSL-SmoM2-YFP/+^*; *PB-MYC/+* mouse model, in which constitutively active SMO-M2 was expressed in GLI1-expressing stromal cells after tamoxifen administration, demonstrating that pathway activity in tumor stromal cells inhibited cancer formation and growth as compared with control mice. The two mouse model studies suggested that Hh pathway activation in tumor stromal cells have tumor-suppressive functions in prostate cancer. In support of this role, two phase I clinical trials tested vismodegib and sonidegib in prostate cancer patients and did not find any clinical benefit despite decreases in GLI levels in the tumors [167,168].

#### 5.1.5. Lung Adenocarcinoma

Lung cancers represent a large portion of cancer diagnoses in the United States and remains the deadliest, with the highest mortality rate of all cancers [145]. Hedgehog signaling pathway activity within the lung is well-documented. The pathway is critical for lung development, particularly for branching morphogenesis, the airways, and alveolar development [169,170,171,172,173,174]. SHH is secreted by the endoderm of the developing lung to activate the pathway in the mesoderm that, in turn, coordinates the expression and secretion of other factors such as WNT2/2b and BMP4 back to the endoderm.

Genetic loss of SHH in the *Kras^LSL-G12D/+^*; *Trp53^fl/fl^* autochthonous mouse model of lung adenocarcinoma after nasal inhalation of adenovirus-cre did not affect tumor growth nor mouse survival [80]. In contrast, treatment of *Kras^LSL-G12D/+^*; *Trp53^fl/fl^* mice with the anti-SHH/IHH blocking antibody, 5E1 [175], early in the tumorigenic process caused increased tumor burden and metastasis, an increase in poorly differentiated tumor histology, and worse survival. IHH, rather than SHH, was demonstrated to be the critical tumor-suppressive Hh ligand through in vivo CRISPR deletion of IHH that recapitulated the phenotype of 5E1 treatment. IHH ligand was secreted from transformed lung epithelia and activated the pathway in lung fibroblasts in a paracrine manner. Loss of stromal Hh pathway activity resulted in decreased angiogenesis and increased reactive oxygen species (ROS). Treatment of *Kras^LSL-G12D/+^*; *Trp53^fl/fl^* mice with 5E1 and the ROS-scavenger, *N*-acetyl cysteine, improved survival rates and inhibited tumor growth compared to 5E1 and vehicle control treated mice, suggesting that one mechanism by which stromal Hh pathway activation restrains early lung adenocarcinoma growth is by supporting angiogenesis to minimize ROS production [80].

### 5.2. Tumor–Promoting Action of Ligand-Dependent Canonical Hh Signaling Pathway

Ligand-dependent canonical Hh pathway activity has now been shown to be tumor suppressive in many of the tumor types that were previously thought to be driven by pathway activity, as noted above. However, tumor-promoting roles of canonical pathway activation has been explored in acute myeloid leukemia and small cell lung cancers, as described below.

#### 5.2.1. Acute Myeloid Leukemia

Acute myeloid leukemia (AML) [176] is an aggressive hematopoietic malignancy characterized by the infiltration and expansion of clonal, abnormally differentiated myeloid cells into the bone marrow, blood, and other organs [177]. Except for the subset of acute promyelocytic leukemia, the primary treatment strategy of cytarabine (Ara-C) and anthracycline induction followed by cytarabine consolidation chemotherapy has not changed in several decades. Patients who are unfit for cytotoxic therapy can be treated with the hypomethylating agent, venetoclax [177].

The precise role of the Hh signaling pathway in AML tumorigenesis is unclear. Hh pathway components SMO and GLI1/2 have all been considered as targets for the treatment of AML, with GLI1/2 expression especially denoting poor prognosis [176,178,179]. Furthermore, crosstalk between the Hh signaling pathway and receptor tyrosine kinase (RTK) pathways has been detected in AML, resulting in increased expression of both FLT3 and GLI2, the combination of which hastens AML proliferation [176,180]. However, results of clinical trials with the SMO antagonist, glasdegib (Pfizer, [181]), as a single agent were underwhelming [182,183]. Induction of the Hh signaling pathway is correlated with therapy resistance in AML. In AML resistant to cytarabine and ribavirin, GLI1 was identified as a driver of UDP glucuronosyltransferase (UGT1A) that modified ribavirin and cytarabine through glucuronidation and diminished their efficacy, leading to drug resistance. Treatment with vismodegib or knock-down of *GLI1* mRNA in combination with ribavirin or cytarabine inhibited the growth of drug-resistant AML cells [70]. The precise mechanism of GLI activation in drug resistance is unclear as drug-resistant cells have diminished levels of PTCH1 but also express SHH and IHH [70]. Upregulation of GLI1 has also been associated with resistance to radiotherapy; treatment with sonidegib, a SMO antagonist, sensitized radioresistant AML cells to radiation in cell culture and in vivo experiments [71]. The sensitivity of drug- and radiation-resistant AML cells to SMO antagonists suggests that the resistance mechanisms are SMO-dependent.

In a phase II trial of AML or high myelodysplastic syndrome (MDS) patients who were at least 75 years old or who could not tolerate first-line intensive chemotherapy, treatment with glasdegib and low dose cytarabine significantly improved overall survival compared to low dose cytarabine alone [184]. Based on these results, glasdegib in combination with low dose cytarabine was FDA-approved for newly diagnosed AML patients 75 years or older or those patients that cannot tolerate intensive chemotherapy [72]. Numerous other trials using glasdegib or sonidegib in combination with other therapies for AML and MDS are ongoing [185].

#### 5.2.2. Small Cell Lung Cancer

Small cell lung cancer (SCLC) is a highly aggressive, smoking associated, neuroendocrine neoplasm that is mostly diagnosed in the metastatic setting [186]. The first connection between SCLC and the Hh signaling pathway [67] was the identification of a subset of SCLC cell lines that expressed SHH and GLI1 and whose growth in culture and in vivo xenografts were inhibited by the SMO antagonist, cyclopamine [99,187,188]. A subsequent study noted that primary SCLC samples had higher frequency of pathway activity as measured by GLI1 expression than SCLC cell lines [189]. In the autochthonous *Rb^fl/fl^*; *Trp53^fl/fl^* mouse model of SCLC [190], expression of the constitutively active mutant SMO-M2 in tumor cells led to increased tumor number and size [68]. Conversely, loss of SMO in the tumor cells led to significant decreases in tumor number and size. Treatment of an SCLC patient-derived xenograft (PDX) cell line with recombinant rSHH or viral infection with SMO-M2 accelerated cell proliferation in culture, whereas pharmacologic inhibition, genetic knockdown, or depletion of SHH/IHH by a blocking antibody inhibited tumor cell proliferation. Treatment of SCLC PDX xenografts with the SMO antagonist, sonidegib, alone mildly inhibited tumor growth. However, the addition of sonidegib after treatment with carboplatin and etoposide chemotherapies significantly delayed the onset and rate of tumor growth compared to the chemotherapy combination alone [68]. Moreover, over-expression or loss of SHH in *Tp53^fl/fl^*; *Rb1^fl/fl^* mouse model led to heightened or diminished tumor growth, respectively [69]. Taken together, these results suggest that the Hh signaling pathway acts to promote tumor growth in an autocrine Hh-ligand dependent oncogenic manner.

Two clinical trials exploring treatment of extensive-stage SCLC with the combination of platinum and etoposide chemotherapy with SMO inhibitors have been reported. A small phase I clinical trial tested the combination of sonidegib with etoposide and cisplatin in newly diagnosed late-stage small cell lung cancer patients [191]. The trial identified the maximum tolerated dose of sonidegib for the combination. Seventy-nine percent (11/15) of the patients were noted to have a partial response [191]. Despite these promising results, a large phase II trial combining the SMO antagonist, vismodegib, with etoposide and cisplatin concurrently and then as maintenance therapy to treat extensive stage SCLC patients in the first line setting showed no overall- or progression-free survival benefit over cisplatin and etoposide alone [192]. In light of the preclinical data described above, the reason for the lack of clinical benefit from vismodegib is unclear and questions remain. Further clinical development of sonidegib and vismodegib for SCLC therapy has been halted.

## 6. Cancers with Noncanonical Activation of GLI Transcription Factors

Here, we define noncanonical GLI activation as that which is independent of SMO that often requires the influence of other intracellular signaling pathways. Whereas the canonical Hh signaling pathway proceeds through the cascade of Hh ligand-PTCH-SMO-GLI, noncanonical GLI activation circumvents this route. Often, noncanonical GLI activation relies on a complex crosstalk between GLI and other signaling pathways, such as MAPK or PI3K-mTOR, that activate GLI transcription of Hh target genes [193] (Figure 2). A summary of cancers with canonical activation of the Hh signaling pathway is listed in Table 2.

### 6.1. Esophageal Cancer

Esophageal carcinoma consists primarily of squamous cell (ESCC) and adenocarcinoma (EAC). ESCC primarily arises in the background of smoking and alcohol use in the proximal and mid-esophagus. In contrast, EAC arises from a background of obesity, gastroesophageal reflux that leads to Barrett’s esophagus with intestinal metaplasia, and smoking, and occurs in the distal esophagus and gastro-esophageal (GE) junction. Increased expression of Hh pathway components, PTCH1, GLI2, and SHH, was detected in ESCC, EAC, and cancerous precursor lesions, Barrett’s metaplasia and squamous dysplasia, which lead to ESCC and EAC, respectively [205]. Both canonical and noncanonical activation of GLI1 was reported in esophageal adenocarcinoma [194]. TNF-α selectively stimulates GLI1 and promotes GLI1 nuclear localization via mTOR phosphorylation of S6K1 in human esophageal adenocarcinoma cell lines, independent of SMO.

However, addition of exogenous SHH to the esophageal adenocarcinoma cell lines increased GLI activity and the combination of vismodegib and everolimus (mTOR antagonist) suppressed tumor growth more than either drug alone in vivo [194]. These results suggest that both canonical and noncanonical activation of GLI1 activation is critical for esophageal adenocarcinoma growth.

### 6.2. Glioblastoma

Glioblastomas (grade IV gliomas) are highly aggressive primary brain tumors which account for nearly half of all primary malignant brain tumor diagnoses [206]. GLI1, a transcriptional effector of the Hedgehog pathway, was first identified to be amplified in malignant glioma [207] and then characterized as a zinc finger transcription factor [42,44]. In the years since, the influence of GLI proteins on gliomas has been elucidated further. Paracrine Hh pathway activity has been reported in progenitor cells of grade 2 and 3 gliomas but not de novo glioblastomas [208]. SHH was expressed in cells expressing the neuronal marker, NeuN, with pathway response in OLIG2+ progenitor cells as evidenced by PTCH1 and GLI1 expression [208]. Expression of FOXM1B, the predominant isoform of FOXM1 expressed in human gliomas, strongly correlated with poor prognosis and was regulated by GLI1 [209]. Insulin receptor substrate I (IRSI) in glioma stem cells (GSCs) was determined to be a GLI1 transcriptional target; furthermore, GLI1 inhibition was found to antagonize IRSI-mediated MAPK pathway activation and obstruct IGF-I mediated cell survival in GSCs [210]. A crosstalk between SHH and PI3K-mTOR pathways was identified, which promoted tumor proliferation and survivability in PTEN-deficient glioblastomas [195]. PTEN-deficient glioblastomas expressed higher levels of *GLI1* and *GLI2* mRNA compared PTEN-expressing glioblastomas. Knockdown of PTEN by siRNA increased *GLI1* and *GLI2* mRNA transcription in PTEN-expressing glioblastomas, suggesting that the PI3K pathway regulated GLI1 and GLI2 expression. Also, *GLI1* mRNA transcription was decreased with the SMO inhibitor, sonidegib, in PTEN-deficient glioblastoma. The combination of sonidegib and buparlisib (BKM120, [211,212]), a pan-PI3K antagonist, inhibited PTEN-deficient glioblastoma growth in neurosphere culture assays and in vivo, whereas the drugs as single agents had little effect [195]. Thus, in PTEN-deficient glioblastomas, GLI1 expression is driven by both SMO-dependent canonical Hh and PI3K pathways.

### 6.3. Lung Squamous Cell Carcinoma

Non-small cell lung cancer (NSCLC) accounts for approximately 85% of lung cancer diagnoses [145]. NSCLC is classified into two primary subtypes: lung adenocarcinoma (LAC) and lung squamous cell carcinoma (LSCC) [213]. Of these two subtypes, LSCC has been connected with noncanonical activation of the Hh signaling pathway. The relationship was first described in 1997, when SHH was detected in LSCC cell lines and tumors but not in the healthy lung tissue of the same LSCC patients [214]. More recently, GLI1 was identified as a key driver of *PIK3CA* amplified LSCC growth [196]. GLI1 expression was regulated by PI3K pathway and independent of SMO activation or inhibition. In contrast to the cell culture studies, treatment with single agent BKM120, a pan-PI3K antagonist, or arsenic trioxide [215], as a GLI antagonist, did not inhibit LSCC tumor growth or GLI1 expression in vivo, whereas combination treatment demonstrated tumor regression and inhibition of GLI1 expression [196].

### 6.4. Malignant Rhabdoid Tumor

Malignant rhabdoid tumors (MRT) are rare but aggressive neoplasms which arise in extracranial soft tissues, most commonly in that of the kidneys. The tumors are typically found in young children and infants but can occur in older individuals [216]. The incidence of MRT has been connected to a deletion of chromosome 22q11.2, the locus of the gene which encodes *SNF5*, a core component of the SWI/SNF complex [217]. The SWI/SNF complex is an epigenetic complex which regulates gene expression through ATP-mediated chromatin remodeling [218]. SNF5 was identified as a key regulator of GLI1 activity by inhibiting GLI1 expression through binding of *GLI1* promoter regions and presumably modifying the chromatin accessibility of *GLI1* promoters [197]. Loss of SNF5 in MRTs led to noncanonical activation of GLI1, both in cell culture and in vivo, that was reversible by exogenous expression of SNF5. Genetic knockdown and pharmacologic inhibition of GLI1 with HPI-1 [219] suppressed MRT growth in cell culture and in vivo, suggesting that GLI1 is critical for MRT growth and progression [197]. Thus, loss of SNF5 in MRT leads to noncanonical activation of GLI1 to drive MRT growth.

### 6.5. Ewing Sarcoma

Ewing sarcoma is an aggressive, mesenchyme-derived tumor that usually arises in the long bones of extremities, bones of the pelvis, and spine, and is seen is most commonly in pediatric patients [220,221]. One of the most characteristic attributes of Ewing sarcoma is the gene fusion of *EWRS1* with various members of the ETS family of transcription factors, of which the *EWRS1-FLI1* fusion accounts for 85% of Ewing sarcomas [221]. GLI1 was found to be overexpressed in Ewing Sarcoma cell lines, regulated by EWS-FLI1 fusion protein independent of canonical Hh signaling pathway, as evidenced by lack of GLI1 response to exogenous SHH and cyclopamine treatment, and critical for Ewing Sarcoma proliferation [198,199]. The EWS-FLI1 oncoprotein binds to the *GLI1* gene promoter regions to initiate its transcription [200]. Treatment of Ewing Sarcoma cell lines with the GLI antagonists, GANT58 [222] and arsenic trioxide [215], inhibited tumor growth in cell culture [199] and in vivo xenografts [201]. These studies indicate that EWS-FLI1 activates GLI1 independently of the Hh signaling pathway to drive tumor growth.

### 6.6. Vismodegib-Resistant Basal Cell Carcinoma

Treatment with SMO antagonists for unresectable or metastatic BCC is now well established [93,94]. However, resistance to SMO antagonists eventually occurs [223]. We have discussed resistance due to mutations in SMO and other pathway components as one mechanism of drug resistance previously (see “Basal Cell Carcinoma”). Atypical protein kinase C ι/λ (aPKC-ι/λ) was identified as a regulator of GLI in a proteomic screen to identify binding partners of Missing in Metastasis (MIM), a scaffolding protein that potentiates GLI activation downstream of SMO [224]. aPKC-ι/λ modulated Hh pathway activity, ciliogenesis, and was overexpressed in vismodegib-resistant BCC [202]. aPKC-ι/λ phosphorylates GLI1 for activation. Genetic loss of pharmacologic inhibition of aPKC-ι/λ via a myristoylated aPKC peptide inhibitor down-regulated Hh pathway activity and vismodegib-resistant BCC growth [202]. Genomic analysis of human and murine vismodegib-resistant BCC identified serum response factor (SRF) and its coactivator megakaryoblastic leukemia 1 (MKL1) as noncanonical potentiators of GLI1 activation [203]. SRF bound to GLI1 as a transcriptional cofactor for the transcription of a subset of GLI1 target genes. Furthermore, treatment with a MKL1 inhibitor, CCG-1423, reduced *GLI1* expression and decreased tumor burden in drug-resistant BCC cells in cell culture and in vivo [203].

### 6.7. Rhabdomyosarcoma and Lung Adenocarcinoma

In mouse embryonic fibroblasts (MEFs), rhabdomyosarcoma and lung adenocarcinoma cells, MKL1 partnered with Jumonji domain-containing histone demethylase 1A (JMJD1A) to stabilize GLI1 [204]. Treatment of rhabdomyosarcoma and lung adenocarcinoma cells with the a JMJD-antagonist, JIB-04 [225], inhibited tumor cell growth, decreased Hh-target gene expression, and induced GLI1 degradation [204]. DYRK1A indirectly inhibits the noncanonical activation of GLI1 by MKL1-JMJD1A complex via inhibition of ABLIM1/2, whose activity permits the shuttling of MKL1-JMJD1A complex into the nucleus [204].

## 7. Conclusions

Initiated by the predilection of Basal Cell Nevus Syndrome patients towards certain cancers, our comprehension of the Hh signaling pathway in cancer has evolved substantially over the past two and a half decades. Initially, the concept was straightforward-mutations in pathway components activated the pathway to drive cancer growth. Our current understanding is more robust and nuanced with new modes of pathway activation that drive distinct phenotypes. Canonical activation of the pathway via genetic aberrations in pathway components and noncanonical activation of GLI1 and GLI2 transcription factors in tumor cells are oncogenic. In contrast, ligand-dependent canonical pathway activation in stromal cells, particularly fibroblasts, is tumor suppressive (Figure 3). The oncogenic role of canonical ligand-dependent pathway activation is unclear. Preclinical evidence points to such an oncogenic role in myeloid malignancies and SCLC. However, subsequent clinical trials utilizing SMO antagonists in these cancers have been mostly disappointing, with one exception, as noted earlier in the discussion of AML.

The distinct modes of pathway activity in cancers offer new avenues for scientific discovery and therapeutic strategies. SMO antagonists are now well established for the treatment of BCCs and by extension, those tumors driven by loss of function PTCH mutations. However, for mutations in SMO and other downstream pathway components, new therapies are required. For tumors with such mutations or those with noncanonical activation of GLI1/2, research for direct and indirect inhibitors of GLI1/2 is an active area of study (reviewed in [226]). Currently, the direct GLI1/2 inhibitors are for research use only. The precise mechanism by which arsenic trioxide, which is FDA approved for acute promyelocytic leukemia, inhibits GLI1/2 [215] is not known. Many of the indirect inhibitors are in clinical use or being tested in trials and may be considered for clinical testing as GLI1/2 antagonists. Combinatorial inhibition of upstream proteins that drive GLI1/2 and inhibition of GLI1/2 may prove to be a potent therapeutic strategy for cancers with noncanonical activation of GLI1/2. Such dual inhibition strategies have precedence. Combination BRAF and MEK inhibition for mutant BRAF-V600E lung cancer and melanoma have been shown to be effective in preclinical studies [196] and is now standard in clinical practice [227,228]. Strategies to extend the tumor-suppressive functions of canonical pathway activation in stroma, either through activation of the positive regulators of Hh ligands or agonism of downstream targets such as BMPs [74], may be best in early-stage cancers as the tumor-suppressive effects of stromal pathway activation are most potent in early tumorigenesis. Clinical benefit derived from such therapeutic strategies will be the ultimate validation of our maturing knowledge of the Hh signaling pathway in cancers.

## Figures and Tables

**Figure 1 cells-11-02523-f001:**
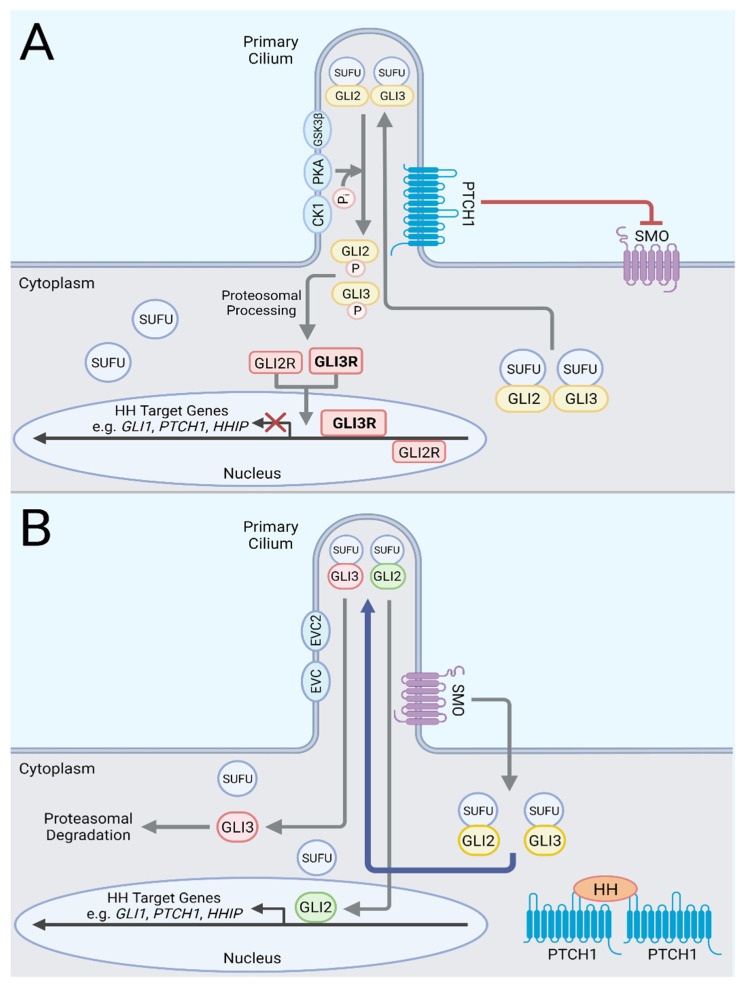
A Simplified Schematic of the Canonical Hedgehog Signaling Pathway. (**A**) Pathway Off. In the absence of a Hh ligand, PTCH inhibits SMO. GLI2 and GLI3, bound to SUFU, traffic to the primary cilia. PKA, CK1, and GSK3β phosphorylate the GLI factors for proteolytic cleavage to their repressor forms, which translocate to the nucleus to inhibit the transcription of Hh pathway target genes. GLI3R (bold) is the major transcriptional suppressor. (**B**) Pathway On. A Hh ligand binds to PTCH to relieves its inhibition of SMO. With SMO activation, SUFU bound to GLI translocates to the primary cilium. EVC and EVC2 promote SUFU dissociation from GLI factors. GLI2, the primary activating GLI factor, then translocates to the nucleus to initiate transcription of Hh pathway target genes. GLI3 undergoes proteasomal degradation. Created with BioRender.com.

**Figure 2 cells-11-02523-f002:**
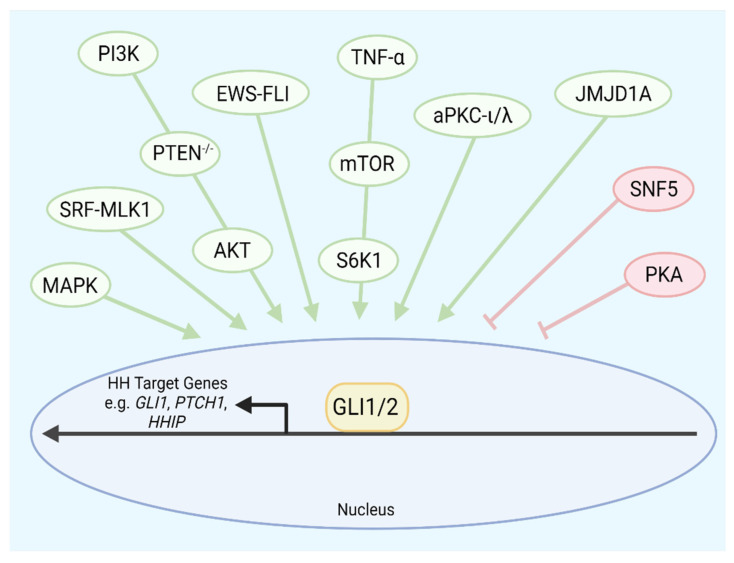
A simplified schematic of regulators of noncanonical GLI activation. Positive (green) and negative (red) regulators of the GLI1 and GLI2 independent of the canonical Hh pathway are shown. Crosstalk between each of these modulators and GLI1/2 bypasses the canonical Hh ligand to PTCH to SMO cascade. Abbreviations: MAPK, mitogen-activated protein kinase; SRF-MKL1, serum response factor-megakaryoblastic leukemia 1; PI3K, phosphoinositide-3-kinase; PTEN, phosphatase and tensin homolog; EWS-FLI, Ewing Sarcoma-Friend Leukemia Integration 1; TNF-α, Tumor necrosis factor α; mTOR, mammalian target of rapamycin; S6K1, S6 kinase beta-1; aPKC-ι/λ, atypical protein kinase C ι/λ; JMJD1A, Jumonji domain-containing histone demethylase 1A; PKA, protein kinase A. Created with BioRender.com.

**Figure 3 cells-11-02523-f003:**
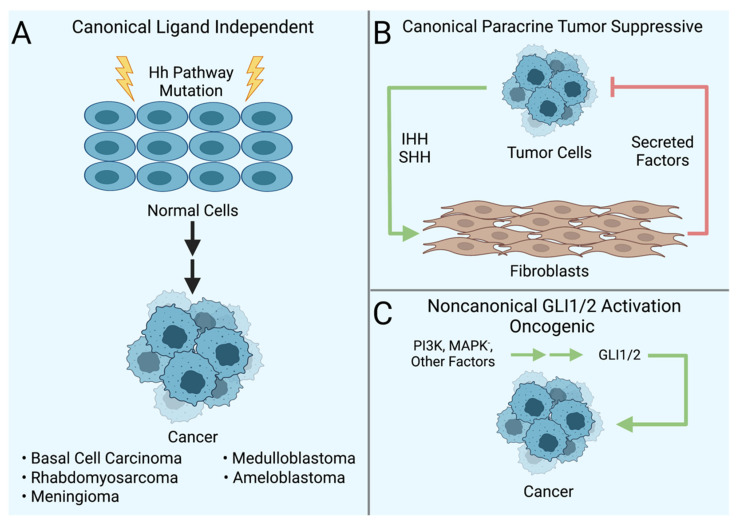
Modes of Hedgehog pathway activity in cancers. (**A**) Canonical ligand-independent activation occurs when mutations in pathway components, e.g., *PTCH1* or *SMO*, cause aberrant activation of the pathway in cancer cells. (**B**) Tumor-suppressive activity of ligand-dependent canonical Hh signaling pathway occurs when Hh ligands produced by tumor cells activate the pathway in neighboring fibroblasts that, in turn, secrete factors, such as BMPs, back to the tumor to inhibit tumor cell growth. (**C**) Noncanonical oncogenic activation of GLI1 and GLI2 transcription factors in tumor cells does not rely on SMO activation. Rather, it arises from the crosstalk between other signaling pathways and GLI1 and GLI2 within the tumor cell, resulting in the activation of GLI1 and GLI2 and the transcription of Hh pathway target genes. Created with BioRender.com.

**Table 1 cells-11-02523-t001:** List of cancers with canonical Hh pathway activation.

Cancer	Hedgehog Signaling Type	Cause of Pathway Activation	Reference
Basal Cell Carcinoma	Ligand-Independent Canonical Oncogenic	Primarily inactivating PTCH1 mutations, secondarily SMO activating mutations (~10%)	[13,14,46,47,48,49,50]
Medulloblastoma	Ligand-Independent Canonical Oncogenic	Loss of function mutations in *PTCH1* and *SUFU*, gain of function mutations in *SMO*	[51,52,53,54,55,56,57,58]
Rhabdomyosarcoma/Rhabdomyoma	Ligand-Independent Canonical Oncogenic	Loss of function mutations in *PTCH1* and activating mutations in *SMO*	[55,59,60,61]
Ameloblastoma	Ligand-Independent Canonical Oncogenic	*SMO* activating mutations	[62]
Meningioma	Ligand-Independent Canonical Oncogenic	*SMO* and *SUFU* mutations	[63,64,65,66]
Small Cell Lung Cancer	Ligand-Dependent Canonical Oncogenic	Overexpression and loss of SHH and loss of SMO in autochthonous tumor cells modulate tumor growth	[67,68,69]
Acute Myeloid Leukemia	Ligand-Dependent Canonical Oncogenic	GLI1 upregulation in chemotherapy- and radiation-resistant AML. SMO antagonists re-sensitize cells to therapy. SMO antagonism with low dose cytarabine improves overall survival in older AML patients. Source of Hh ligand is unknown currently.	[70,71,72]
Bladder	Ligand-Dependent Canonical Tumor Suppressive	Stromal Hh pathway activation by SHH from tumor epithelia inhibits tumor growth by secretion of BMP4 and BMP5 differentiation factors to tumor epithelia	[73,74]
Pancreas	Ligand-Dependent Canonical Tumor Suppressive	Stromal Hh pathway activation by SHH from tumor epithelia inhibits tumor growth, metastases, and increases formation of well differentiated pancreas cancers	[75,76]
Colon	Ligand-Dependent Canonical Tumor Suppressive	Deletion of *IHH* from intestinal epithelia increased tumor burden. Genetic loss or pharmacologic inhibition of stromal SMO increased tumor burden	[77,78]
Prostate	Ligand-Dependent Canonical Tumor Suppressive	IHH secretion from prostate tumor epithelia activates stromal Hh pathway to inhibit tumor growth	[79]
Lung Adenocarcinoma	Ligand-Dependent Canonical Tumor Suppressive	IHH secretion from lung tumor epithelia activates stromal Hh pathway to inhibit tumor growth and metastasis	[80]

**Table 2 cells-11-02523-t002:** List of cancers with noncanonical GLI transcription factor activation.

Cancer	Hedgehog Signaling Type	Cause of Pathway Activation	Reference
Esophageal Adenocarcinoma	Noncanonical GLI Activation Oncogenic	TNF-α-mediated activation of mTOR promotes GLI1 activity	[194]
Glioblastoma	Noncanonical GLI Activation Oncogenic	PI3K-mTOR activity increases *GLI1* and *GLI2* expression	[195]
Lung Squamous Cell Carcinoma	Noncanonical GLI Activation Oncogenic	Increased PI3K activity from *PIK3CA* amplification drives GLI1 expression and activity	[196]
Malignant Rhabdoid Tumor	Noncanonical GLI Activation Oncogenic	Loss of SNF5 leads to elevated GLI1 expression and activity	[197]
Ewing-Sarcoma	Noncanonical GLI Activation Oncogenic	EWS-FLI1 complex binds to the *GLI1* gene promoter region to induce *GLI1* transcription	[198,199,200,201]
Vismodegib-resistant Basal Cell Carcinoma	Noncanonical GLI Activation Oncogenic	aPKC-ι/λ and MIM promote GLI activation downstream of SMO. SRF and co-regulator MKL1 potentiate GLI1 activation	[202,203]
Rhabdomyosarcoma, Lung Adenocarcinoma	Noncanonical GLI Activation Oncogenic	JMJD1A and co-regulator MKL1 stabilize GLI1	[204]

## Data Availability

Not applicable.

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
