# Peer review of "Canonical Hedgehog Pathway and Noncanonical GLI Transcription Factor Activation in Cancer"

_cells, 2022, doi:10.3390/cells11162523_

Round 1
Reviewer 1 Report
The review provides a very nice overview of the contributions of aberrant Hedgehog pathway activation in cancer. The authors have focused on “non-canonical” signalling which is a novel and interesting angle to discuss.
My single concern with the paper is that the authors discuss only the transcriptional non-canonical Hedgehog pathway and largely ignore the studies on G-protein signalling and cytoskeletal events (work from for instance the Roelink, Charron, and Riobo labs, but there are others as well). Indeed, they mention “we will focus on Hh ligand-independent and SMO-independent activation of GLI transcription factors in our discussion of noncanonical activity of the pathway in cancers.” but this is a very narrow view of the field.
Author Response
Response to Reviewer 1
We thank Reviewer 2 for their kind and insightful comments. We agree that the other modes of noncanonical Hh pathway, such as those noted by Reviewer 1, are important. However, these other modes of noncanonical signaling have been reported primarily in normal/benign tissue and developmental contexts with few reports in cancer contexts. In contrast, there is a growing body of cancer literature on noncanonical activation and upregulation of GLI1/2 as we have noted in our review. Given this, we have modified the title, abstract, main text, Table 2 and Fig. 3 of the review to be more precise that we are reviewing canonical Hh pathway activation and noncanonical activation of GLI transcription factors in cancers. We also acknowledge in the main text that other modes of noncanonical Hh signaling have been reported primarily in benign and developmental contexts and thus, will focus on noncanonical GLI activation in cancers.
Reviewer 2 Report
This is an excellent review on the Sonic Hedgehog pathway in cancer. I could not find any relevant flaws. The text is very well written, the figures are great and the literature is broadly cited. The authors address the role of SHH in cancer linking it to its role in development in a balanced way. The relevant types of cancer where SHH plays a role are discussed. This review can be published as it is.
Author Response
Response to Reviewer 2
We thank Reviewer 2 for their kind comments regarding our review manuscript.